# Variation in Road Dust Heavy Metal Concentration, Pollution, and Health Risk with Distance from the Factories in a City–Industry Integration Area, China

**DOI:** 10.3390/ijerph192114562

**Published:** 2022-11-06

**Authors:** Yinan Chen, Zhiqiang Hu, He Bai, Wei Shen

**Affiliations:** 1Key Research Institute of Yellow River Civilization and Sustainable Development & Collaborative Innovation Center on Yellow River Civilization Jointly Built by Henan Province and Ministry of Education, Henan University, Kaifeng 475001, China; 2College of Resources and Environment, Henan Agricultural University, Zhengzhou 450018, China; 3School of Environmental and Chemical Engineering, Shanghai University, Shanghai 200444, China; 4The College of Environment and Planning, Henan University, Kaifeng 475001, China

**Keywords:** road dust, heavy metals, spatial variation, pollution, health risk, manufacturing factory

## Abstract

Road dust samples around three typical factories, F1, F2, and F3, in the National Zhengzhou Economic and Technology Development Zone (ZETZ), China, were collected to study the variation in heavy metal concentration (As, Cd, Cr, Cu, Hg, Ni, Pb, and Zn), pollution, and health risk with distance from the factories. The results indicated that the concentrations of all the elements near F1 were higher than near both F2 and F3. Apart from Co, Mn, and Cu in some dust samples, all the element concentrations were higher than the corresponding background values (BCs), to varying degrees. The spatial distributions of the heavy metals surrounding the factories followed the normal distribution. The peak values of element concentrations occurred at 300~400 m away from the factories, except for Hg, which continued increasing more than 500 m away from the factories. The fluctuation curves of the pollution load index value calculated according to the BCs for F1, F2, and F3 all had two peaks, a “small peak” and a “large peak”, appearing at about 30 m and 300 m, respectively. For the hazard index and the total carcinogenic risk, the peak values all appeared at 400 m, with the curves following the normal distribution. Exposure to road dust containing non-carcinogenic and carcinogenic elements around F1 was greater than around F2 or F3. In conclusion, our results provide a reference for pursuing effective prevention of dust heavy metal pollution around modern manufacturing factories.

## 1. Introduction

Road dust refers to solid particles on the road surface with a particle size of less than 20 mesh (<0.92 mm) [1]. These can adsorb more heavy metal pollutants due to their large specific surface area, making them sensitive to pollution sources [2,3,4]. A large number of studies [5,6] have shown that, owing to human activities, the concentration of heavy metals in road dust far exceeds original levels, thereby leading to potential health risks for local people through inhalation, ingestion, and dermal contact [7,8]. Especially in densely populated cities, more people are at risk of exposure to road dust heavy metals [9,10]. For these reasons, studies of the variation in road dust heavy metal distribution, pollution, and health risk with distance from factories could contribute greatly to the assessment of public health.

City–industry integration, which requires the spatial integration of industry and urban functions, has been an important development pathway for new-type urbanization in China in recent years [11]. In the past 10–20 y, urbanization and industrialization in these areas have developed at an unprecedented rate. Previous studies demonstrate that industrial activities always lead to a significant increase in the concentrations of heavy metals in dust [12,13]. Consequently, humans who live and work in the city–industry integration areas, specifically near the factories, may have more potential health risks due to high heavy metal exposure currently and in the future. Existing studies to determine and monitor urban road dust heavy metal pollution levels have been focused on functional areas such as bus stations or parks [14,15], while the studies on the spatial variation of heavy metal concentrations have been focused on the areas around high pollution factories far away from residential areas [12,16]. There have been few studies focusing on the dust heavy metal contamination caused by the modern high-tech manufacturing factories, especially those located in the typical city–industry integration areas with dense populations. Therefore, in this study, the concentrations, diffusion range, and health risks of heavy metals in the dust around such factories were investigated.

The National Zhengzhou Economic and Technology Development Zone (ZETZ), as one of the typical city–industry integration areas, was taken as the study area. Road dust samples were collected from three typical factories, the electrical component manufacturing factory (F1), the automobile production and parts manufacturing factory (F2), and the equipment manufacturing factory (F3) [17]. The objectives of the study were (1) to determine the dust heavy metal concentrations and their distribution, including As, Cd, Co, Cr, Cu, Hg, Mn, Ni, Pb, and Zn; (2) to explore the degree of pollution with heavy metals in road dust by calculating the pollution load index (*PLI*), and measuring its spatial distribution; and (3) to investigate the non-carcinogenic and carcinogenic health risks posed to humans, and their spatial distribution characteristics, by applying the United States Environmental Protection Agency (USEPA) health risk assessment model.

## 2. Materials and Methods

### 2.1. Study Area

The ZETZ, founded in 1993, is located in the southeast suburbs of Zhengzhou city, Henan Province. It occupies an area of approximately 158.7 km^2^, with a permanent resident population of about 342,600. It has a temperate monsoon climate, with a mean annual temperature of 14.3 °C and mean annual rainfall of 640.5 mm, and is mainly covered by the alluvial deposits of the Yellow River (Figure 1).

### 2.2. Sample Collection

In this study, dust samples were collected from the areas surrounding F1, F2, and F3. Nine dust samples were collected in each direction, at distances of 0–30 m, 30–60 m, 60–90 m, 90–150 m, 150–200 m, 200–250 m, 250–300 m, 300–400 m, and 400–500 m from the factory (Figure 2). Sampling points should have been arranged according to the dominant local wind direction, but due to the restrictions regarding the factory surroundings, listed in Appendix A, the number and direction of the sampling transects for each factory were different. Even so, when sampling, we ensured that the road section was completely covered in every sampling point, to try to make it more representative.

A total of 78 road dust samples were collected from the three factories, comprising 18 samples from F1, 18 samples from F2, and 36 samples from F3. All the samples were collected from the surrounding asphalt road using plastic brushes and plastic crucibles, then stored in labeled polyethylene bags until analyzed in the laboratory.

Thirty control soil samples were also collected from eastern virgin regions located more than 3 kilometres from the study area. The mean content of each element was calculated to produce their corresponding background values (BCs). Details of that calculation can be found in the article by Chen et al. [15].

### 2.3. Sample Preparation and Heavy Metal Analysis

Road dust samples were air-dried and then filtered to remove large and thick contaminants such as cigarette butts, leaves, and rocks. The samples were then manually ground by agate mortar and pestle, passed through a nylon sieve of 0.149 mm, and stored in closed polyethylene bags before chemical assay.

For the determination of Cd, Co, Cr, Cu, Mn, Ni, Pb, and Zn concentrations, 0.1000 g samples were weighed, placed into PVC digestion vessels, and digested in an automatic graphite digestion instrument (ST-60, Polytech, Beijing, China) with a mixture solution of concentrated HNO_3_-HF-HClO_4_ [18,19]. The concentrations of the target heavy metals in the dust sample were determined by inductively coupled plasma mass spectrometry (ICP-MS; X-Series II Model, Thermo Fisher Scientific, America) and inductively coupled plasma atomic emission spectrometry (ICP/AES; ICPS-7500, Shimadzu, Japan).

To determine the concentrations of As and Hg, 0.2000 g of sample was placed into a 25 mL colorimetric tube, digested in a water bath with a mixture solution of aqua regia [20,21], and analyzed by atomic fluorescence spectrometry (AFS 830, Titan, China).

### 2.4. Quality Assurance and Quality Control

In the course of the experiment, in order to completely remove the residual heavy metal ions from the experimental equipment, the Teflon digestion tank and all the glass laboratory equipment were immersed in 20% and 10% nitric acid for more than 24 h, respectively, then washed with deionized water 3 times.

Two procedural blanks, two standard reference materials (GBW-07405 for soils, National Research Center for Standards, China), and at least 10% parallel samples were included with each batch. The procedural blanks for all analyses were below the method detection limit (MDL). The recoveries of the elements ranged from 88.06% to 109.47%.

### 2.5. Heavy Metal Pollution Assessment

Potential load index (*PLI*) was used to evaluate the dust heavy metal pollution [22]. The function can be formulated as follows:(1)$${\mathrm{CF}}_{i}=\frac{c_{i}}{c_{n}}$$
(2)$$\mathrm{PLI}=\sqrt[{n}]{{\mathrm{CF}}_{1}\times {\mathrm{CF}}_{2}\times \cdots \times {\mathrm{CF}}_{n}}$$
where *CF_i_* is the pollution index of heavy metal *i*, *c_i_* (mg·kg^−1^) is the concentration of element *i* in dust samples, and *c_n_* is the background value of heavy metal *i*. BCs and China’s tidal soil background value were used for evaluation in this study. Because there lacked the level of “slight pollution” [22], the classification standards of *CF* and *PLI* were adjusted as: no pollution (*CF/PLI* ≤ 1), slight pollution (1 < *CF/PLI* ≤ 2), moderate contamination (2 < *CF* ≤ 3), and heavy pollution (*PLI* ≥ 3) [23].

### 2.6. Health Risk Assessment Model

The health risk assessment model was developed by the USEPA [8] to quantify human health risks (carcinogenic and non-carcinogenic). People were exposed to road dust by three main pathways: ingestion, inhalation, and dermal contact. The estimation of exposure dose for the three pathways was calculated using the following equations: (3)$${ADD}_{\mathrm{ing}}=\frac{c\cdot \mathrm{Ing}R\cdot CF\cdot EF\cdot ED}{BW\cdot AT}$$
(4)$$ADD_{\mathrm{inh}}=\frac{c\cdot \mathrm{Inh}R\cdot EF\cdot ED}{PEF\cdot BW\cdot AT}$$
(5)$$ADD_{\mathrm{dermal}}=\frac{c\cdot SA\cdot CF\cdot SL\cdot ABS\cdot EF\cdot ED}{BW\cdot AT}$$

The potential non-cancer risk was assessed using the hazard quotient (*HQ*) and the hazard Index (*HI*), calculated in Equations (6) and (7), and the total carcinogenic risk (*TCR*) was calculated using Equations (8) and (9):(6)$$HQ_{i}=\sum_{j=1}^{3}\frac{ADD_{ij}}{RfD_{ij}}$$
(7)$$HI=\sum_{i=1}^{9}HQ_{i}$$
(8)$${Risk}_{ij}={ADD}_{ij}\times {SF}_{ij}$$
(9)$$TCR=\sum_{ij}^{3}Risk_{ij}$$
where *RfD* is the reference dose and *SF* is the cancer slope factor. Definitions of the abbreviations, units, and values of each parameter in the formulas are also presented in Appendix A [7,8,24,25,26,27,28,29].

## 3. Results

### 3.1. Characteristics of Dust Heavy Metal Concentrations

The heavy metal concentrations in road dust around the typical factories of ZETZ are presented in Table 1. The average concentrations of As, Cd, Co, Cr, Cu, Hg, Mn, Ni, Pb, and Zn were 6.65, 0.52, 5.74, 41.78, 18.31, 0.034, 371.72, 19.65, 41.62, and 117.13 mg·kg^−1^, which were 1.64, 1.85, 0.74, 1.46, 1.54, 0.92, 1.18, 1.64, 2.74, and 3.59 times the BCs, respectively. Apart from Co and Hg, all the other heavy metals were accumulated in higher concentrations than the BCs. In addition, compared with previous studies, the average road dust heavy metal concentrations in ZETZ was lower than those in big cities such as Beijing [30], Shanghai [31], and Guangzhou [32]. This indicates that although the heavy metal concentrations in road dust were likely affected by the local human activities, their accumulation was not severe.

The average total dust heavy metal concentrations near the three factories followed the order F1 (740.72 mg·kg^−1^) > F3 (590.55 mg·kg^−1^) > F2 (570.85 mg·kg^−1^). For F1, concentrations of As, Co, Cr, Cu, Ni, Pb, and Zn were higher than for both F2 and F3. Cd and Mn concentrations were higher than F2, but lower than F3, whereas Hg was lower than F2, but higher than F3. For all three factories, most of the studied element concentrations in road dust were higher than the BCs. Pb and Zn were the most accumulated elements, especially near F1, where they were 4.26 and 5.33 times higher than the BCs, respectively. Those near F2 and F3 were 1.98 and 2.73 times higher than the BCs, respectively. The high concentrations of heavy metals in road dust around the factories were most likely due to the increase in industrialization and vehicular emissions [33,34].

### 3.2. Spatial Distribution of Dust Heavy Metal Concentrations with Distance from the Three Factories

There were similarities and differences in the spatial variation of dust heavy metal concentrations around F1, F2, and F3 (Figure 3). Generally, the fluctuation of Mn concentrations around these three factories was relatively stable and consistent.

The spatial variability around F1 was the most significant, and the peak values were the most obvious. For F1, F2, and F3, the fluctuation patterns of Pb, Zn, and Cu were similar, which presented as “Decrease (30–60 m)-Increase (60–300 m)-Decrease (300–500 m)”. Most of the peak values appeared at about 300 m away from the factories.

The fluctuation patterns of As, Co, Hg, and Ni at F2 and F3 were relatively gentle, indicating that these heavy metals were less affected by the local industrial activities. Whereas, for F1, the concentrations of these elements in dusts changed significantly. Arsenic slowly increased to the highest value (400 m), and then dropped to the BC (500 m). The peak value of Hg was very likely reached beyond 500 m. Co and Ni fluctuated greatly but without obvious trends and may be affected by multiple surrounding factors.

The gentle fluctuation of Cr and Cd around F1, F2, and F3 was very similar. The peak value of Cd appeared at 250–300 m for all the three factories.

### 3.3. Heavy Metal Pollution Assessment

In order to describe the degree of pollution of heavy metals in dust at different distances from the factories, two sets of evaluation criteria, soil BCs and the National Soil Environmental Standard of Grade II [35], were selected and calculated according to Equation (1). The order of *CF* values for each element, calculated according to the BCs (reflecting the concentration of heavy metals in the dust by factory), were (Table 1):

F1: Zn (5.33) > Pb (4.26) > Hg (2.58) > As (2.52) > Ni (2.40) > Cd (1.89) > Cu (1.78) > Cr (1.69) > Mn (1.17) > Co (0.94).

F2: Zn (2.73) > Pb (1.94) > Cr (1.52) > Cu (1.46) > Hg (1.33) > Cd (1.23) > Ni (1.19) > As (1.14) > Mn (1.13) > Co (0.66).

F3: Zn (2.72) > Cd (2.44) > Pb (2.03) > Hg (1.62) > Cu (1.39) > Ni (1.28) > As (1.27) > Mn (1.23) > Cr (1.17) > Co (0.61).

For the three factories, Zn and Pb were the main sources of pollution, while the accumulation of Mn and Co was weak. There was much variability in the pollution order of other elements, which could be caused by the combined pollution of industrial activities and the surrounding environment.

Due to the lower concentrations of heavy metals in the dust around F2 and F3, the *PLI* values for those two factories were lower than for F1. The mean *PLI* values followed the order of F1 (2.00) > F3 (1.34) > F2 (1.27) and F1 (0.36) > F3 (0.25) > F2 (0.23), calculated according to the BCs and the National Soil Environmental Standard II [35], respectively. From Figure 4, it can be seen that the *PLI* values based on the national secondary standard were low, with little fluctuation, whereas the *PLI* values calculated based on the BCs were all higher than 1, and the spatial fluctuation was more obvious. The *PLI* values of F1, F2, and F3 all showed two peaks, at about 30 m and 300 m. There was a decreasing trend at 30–200 m, and an increasing trend in the range of 200–400 m. Meanwhile, most of curves showed a peak value near 300 m. After 400 m, the *PLI* values showed a gradual decreasing trend. A comparison of the three factories found that the *PLI* value of F1 was greater than 2 in the range of 30 m and 200–400 m away from it, indicating that the factory has a significant impact on the concentration of heavy metals in dust.

### 3.4. Human Health Risk Assessment

The risk assessment model was used to determine both the non-carcinogenic and carcinogenic risk from the dust samples around each factory in adults and children through dermal, ingestion, and inhalation pathways. The non-carcinogenic risks of heavy metals in dust followed the order of F1 (1.17) > F2 (0.71) > F3 (0.70) and F1 (0.02) > F2, F3 (0.01, 0.01) for children and adults, respectively. The results of the non-carcinogenic risk assessment of heavy metals in dust indicated unlikely adverse health effects for adults around the factories, but likely adverse health effects for children in some areas around F1. As found by other studies, children are at higher risk via direct ingestion of dust than adults [6]. In addition, it was determined that both adults and children were most affected by As, Cr, and Pb via the three exposure pathways among the non-carcinogenic heavy metals. The average *HQ* values showed the following decreasing patterns: 

Children: As (3.11 × 10^−1^) > Cr (1.99 × 10^−1^) > Pb (1.64 × 10^−1^) > Mn (1.12 × 10^−1^) > Co (2.71 × 10^−2^) > Hg (1.51 × 10^−2^) > Ni (1.33 × 10^−2^) > Cd (6.87 × 10^−3^) > Cu (6.02 × 10^−3^) > Zn (5.35 × 10^−3^).

Adults: As (4.02 × 10^−2^) > Cr (2.6 × 10^−2^) > Pb (2.10 × 10^−2^) > Mn (1.72 × 10^−2^) > Co (3.55 × 10^−3^) > Ni (1.71 × 10^−3^) > Cd (9.07 × 10^−4^) > Cu (7.72 × 10^−4^) > Zn (6.85 × 10^−4^) > Hg (3.07 × 10^−4^).

Accordingly, the average of the total *HI* was found to be approximately ten times higher in children than adults for each analyzed heavy metal, showing that children are more vulnerable to non-carcinogenic health risks than adults. This may be due to the fact that children are exposed to dust through their habits of playing in the dust with toys and their hand-to-mouth behavior [36]. No health risks were observed from the elements as the *HI* < 1, thereby illustrating no risk and subsequent adverse health effects.

The carcinogenic risks for the three factories were calculated for As, Cd, Co, Cr, and Ni according to possible exposure routes for both children and adults. The average *TCR* values for the observed factories were found in the decreasing order of F1 (1.90 × 10^−5^, 1.00 × 10^−5^) > F3 (9.56 × 10^−6^, 5.07 × 10^−6^) > F2 (8.61 × 10^−6^, 4.59 × 10^−6^) for children and adults, respectively. The average *CR* values were found in the order of As (6.46 × 10^−6^, 1.23 × 10^−5^) > Cr (9.56 × 10^−8^, 5.57 × 10^−8^) > Co (3.04 × 10^−9^, 1.77 × 10^−9^) > Ni (9.02 × 10^−10^, 4.75 × 10^−10^) > Cd (1.60 × 10^−10^, 9.30 × 10^−11^) for children and adults, respectively. It was found that exposure to road dust containing carcinogenic metals around F1 was more dangerous than around F2 and F3. Furthermore, the results indicated that both *CR* and *TCR* for children and adults were far from reaching the threat threshold (1 × 10^−6^, 1 × 10^−4^).

The distributions for average non-carcinogenic and carcinogenic risks of heavy metals in road dust for the three factories are shown in Figure 5. For both children and adults, the peak value of potential health risk appeared at 400 m. The fluctuation of non-carcinogenic risk was greatly affected by As, Pb, and Cr. For the spatial trends of carcinogenic risk, As and Cr were the biggest contributors.

## 4. Discussion

### 4.1. Effects of Factory Type on Dust Heavy Metal Concentrations

Characteristics of dust heavy metal concentrations were directly affected by the factory type [37]. Most of the heavy metal concentrations near F1 were significantly higher than those near F2 and F3 (Figure 6). For F1, computer, communication, mobile phone, consumer electronics, and other such components are the main products. Various heavy metal pollutants could be produced during the processing and manufacturing of circuit boards, batteries, semiconductors, and liquid crystal displays, etc. [38,39]. Although F1 belongs to the high-tech industry, it is still a typical labor-intensive factory. Thus, the emission of pollutants in the production process can be difficult to avoid completely, and could have a serious impact on the surrounding dust [40]. In this study, all the elements were accumulated to a certain degree, but the most obvious pollutants were Zn, Pb, Hg, As, and Ni.

F2 is an automobile manufacturing factory. The primary products are domestic cars, pickup trucks, and auto parts, which means processes such as manufacturing engines, spraying, general assembly, and interior decoration, etc., are common. F3 is an automobile equipment manufacturing factory. The main products are new energy trucks and buses, including the manufacturing of key parts and components, as well as the processes of stamping, welding, coating, and assembly. Both F2 and F3 adopt advanced technologies, such as fully enclosed noise reduction, dust isolation factory building, fully automatic assembly line production, and digital control, which greatly reduce the pollution caused by emissions from traditional production processes. The most highly accumulated pollutants near F2 were Zn, Pb, Cr, Cu, and Hg, and those near F3 were Zn, Cd, Pb, Hg, and Cu, which mainly came from the automobile electroplating process [41,42,43] and the impact of human activities such as traffic and transportation surroundings [44,45,46]. In this study, the production processes, technical advancement, and environmental protection facilities in F3 were better than in F2, which resulted in F3 having a smaller impact on dust heavy metal concentrations.

### 4.2. Factors Affecting the Spatial Distribution of Dust Heavy Metal Concentrations

The discharge intensity of the pollutant is the mostly important factor affecting the special distribution. Many studies have been conducted [47] which reveal that heavy metal concentrations around a pollution source generally obey the normal distribution, which reaches peak value at a certain distance from the pollution source. For example, the peak value of heavy metal content in roadside soil generally occurs within 25–50 m from the subgrade [48]. The main settlement area of the heavy metals discharged from a waste incineration factory was in the range of 1000 m [49,50]. Accordingly, because the emission intensity of different pollution sources varies greatly, the distances for the peak values of heavy metal concentrations around various factories will be different, even for the same element [51,52]. In this study, the general spatial distribution trends of dust heavy metal concentrations were consistent, but the details were different. This could be due to the fact that F1, F2, and F3 are located in the same zone and are subject to the same environmental regulations, but they are different types of factories.

The spatial distribution characteristics of heavy metal concentrations around the pollution source are also influenced by the pollutants’ physical and chemical properties, including volatility, discharged particle size, etc. In this study, the peak value of Hg near F1 appeared far greater than other elements at about 500 m. Fan [39] indicates that when emitted by a coal-fired power factory within 1000 m from the chimney in the downwind direction, the peak value of Hg concentration appeared at 1500 m. These results confirm that the volatility of Hg makes it spread farther than other elements. Meanwhile, the particle sizes of different pollution sources differ, therefore the distance the particles will diffuse in the air will be different, which directly affects the place where the peak value of dust heavy metals occurs [48,53,54].

In addition, it is reasonable to consider the surrounding environment when looking at the factors that influence the distribution of heavy metal concentrations, including nearby human activities, wind direction and speed, and landform [55,56,57]. This could explain the lack of an obvious law of distribution in the Cd and Cu fluctuations around the three factories, which could be due to the impacts of fertilization and spraying of plants in the surrounding green space [58,59]. The finding of the highest average Hg concentration near F3 could be due to that the factory is proximity to the rural business areas which related the heating and catering activities [60].

### 4.3. Effects of Various Heavy Metal Concentrations on Spatial Distribution Characteristics for Pollution and Health Risk

The curves of *PLI* values calculated according to BCs for F1, F2, and F3 all had two peaks, a “small peak” and a “large peak”, that appeared at about 30 m and 300 m, respectively. In the dust around the three factories, Pb and Zn contributed the most to the comprehensive pollution load index (*CF*/*PLI*).

For *HI* and *TCR*, the peak values appeared at 400 m, with a normal distribution curve. The contribution rate of As to the total risk (*HQ*/*HI*) at F1, F2, and F3 was 30.55%, 34.20%, 40.77%, respectively, and its contribution to the total carcinogenic risk (*CR*/*TCR*) was 98.84%, 97.75%, 98.42%, respectively.

The spatial distributions found by the two evaluation methods were different. *PLI* focuses more on the accumulation characteristics of heavy metals based on the BCs, indicating that Pb and Zn in road dust were the most prevalent pollutants surrounding the factories. Whereas *HI* and *TCR* focus more on the toxicity response coefficients of the heavy metals, and the toxicity coefficients of As were found to be much higher than the other heavy metals.

## 5. Conclusions

In this study, the variation in road dust heavy metal concentrations, pollution, and health risk with distance from factories in the city–industry integration area, ZETZ, were investigated. The results were as follows:

Road dust heavy metal concentrations around the electrical component manufacturing factory (F1) were higher than those around the automobile production and parts manufacturing factory (F2) and the transportation equipment manufacturing industry factory (F3). Therefore, it is necessary to strengthen the environmental control of factories such as F1. Factories such as F2 and F3 should focus on the Cd, Cu, and Cr emissions from electroplating. In addition, the significant increase in Pb and Zn content in surface dust due to industrial activities and large traffic volume should be considered.

The *PLI* indicated that exposures to road dust heavy metals near F1 were more dangerous than those near F2 and F3. At present, however, there is no potential risk exposure to dust heavy metals around the three typical factories in the study area. The health risk of F1 was higher than that of F2 and F3, however both non-carcinogenic and carcinogenic risks for children and adults were far from reaching the threat thresholds.

The spatial change curves for the heavy metal concentrations around the factories generally followed a normal distribution, and the peak values mainly occurred at 300~400 m away from the factories. The peak values for *PLI,* which was calculated according to the BCs, appeared at about 30 m and 300 m, respectively. For *HI* and *TCR*, the peak values appeared at 400 m. In general, relevant preventative measures for heavy metal pollution should be strengthened, especially at 300~400 m away from factories.

## Figures and Tables

**Figure 1 ijerph-19-14562-f001:**
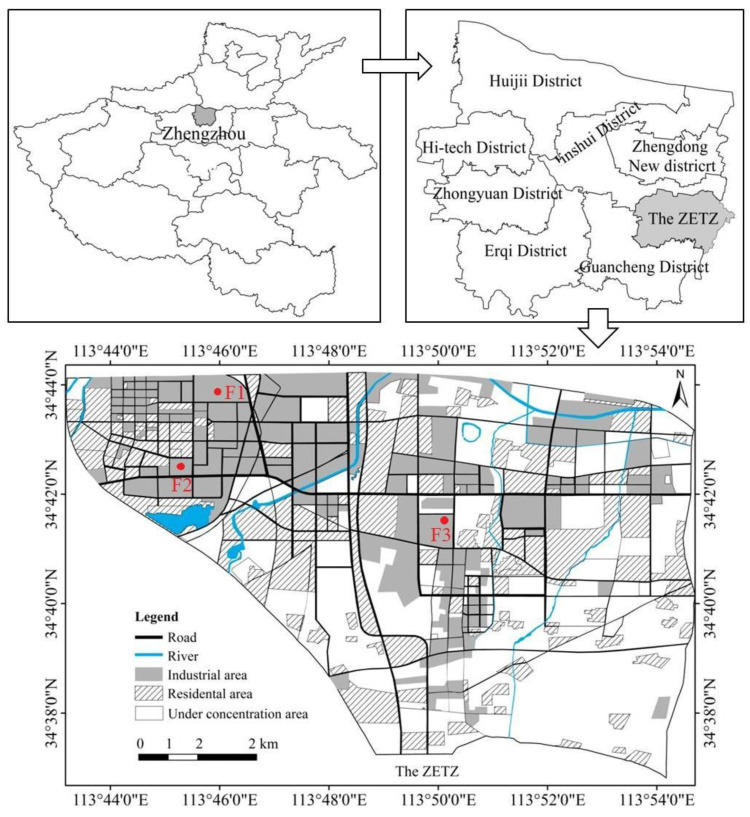
Map of the study area and sample points.

**Figure 2 ijerph-19-14562-f002:**
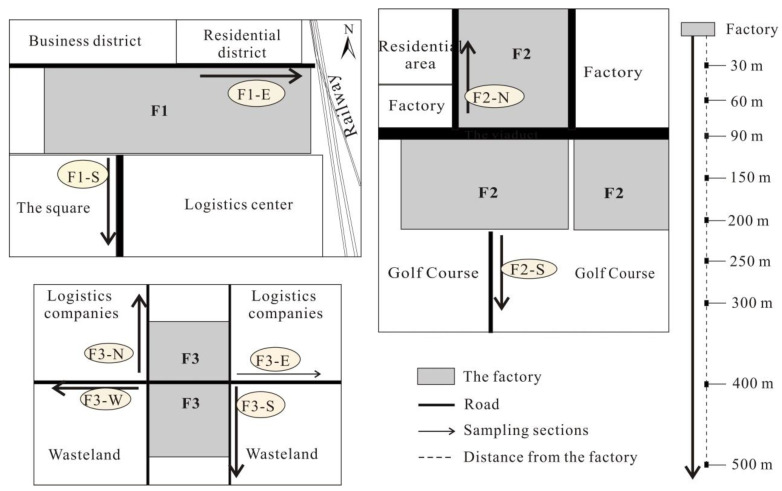
Distribution of the dust sampling sections.

**Figure 3 ijerph-19-14562-f003:**
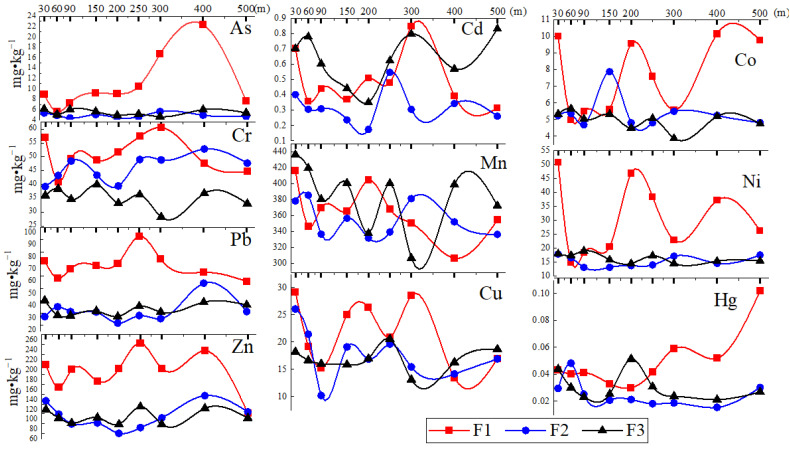
Dust heavy metal concentrations around the three factories.

**Figure 4 ijerph-19-14562-f004:**
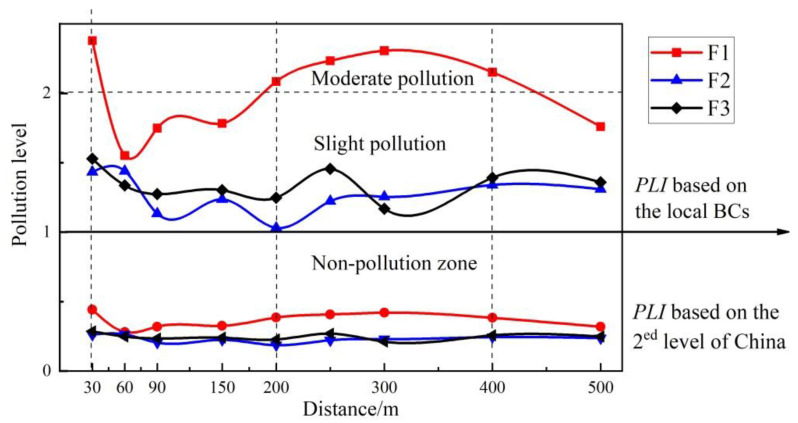
*PLI* values and heavy metal contamination of dust in sampling transects.

**Figure 5 ijerph-19-14562-f005:**
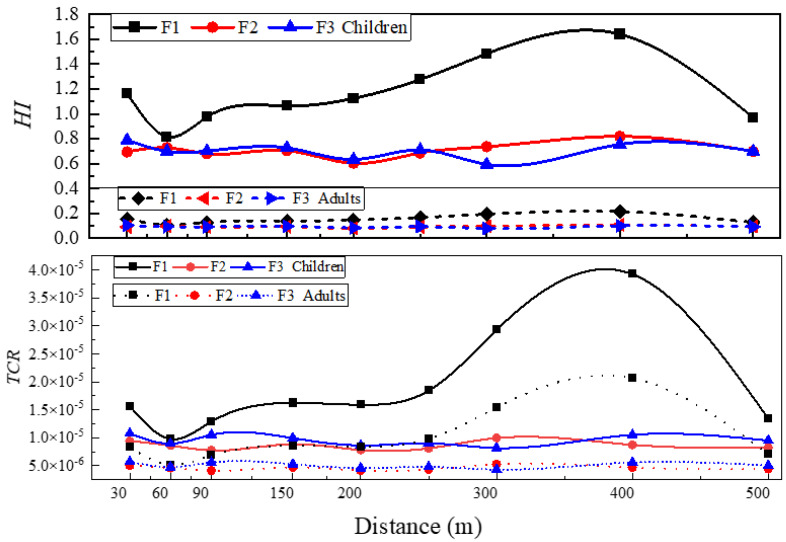
Spatial distribution of non-carcinogenic and carcinogenic risks for the three factories.

**Figure 6 ijerph-19-14562-f006:**
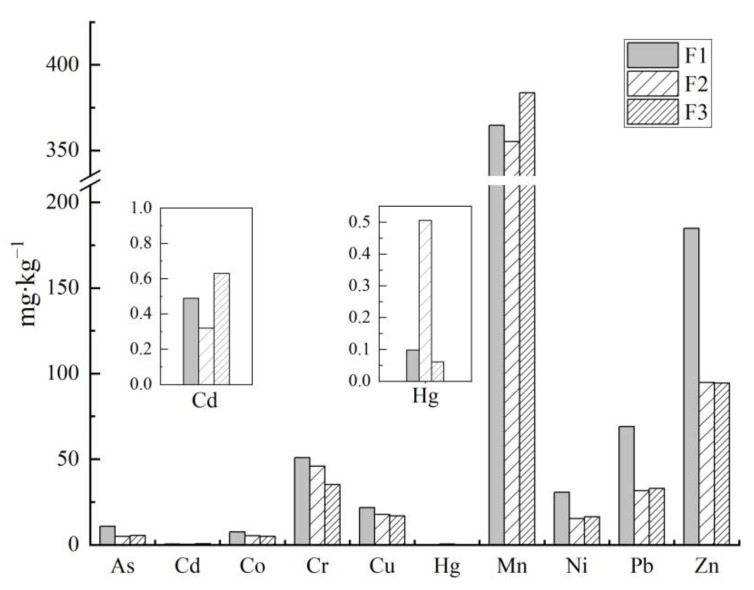
Comparison of heavy metal concentrations for three factories and the BCs.

**Table 1 ijerph-19-14562-t001:** Statistical description of the dust heavy metal concentrations around the three factories.

Items		Heavy Metals Concentrations/mg∙kg^−1^									
		As	Cd	Co	Cr	Cu	Hg	Mn	Ni	Pb	Zn
F1 (*n* = 18)	Min	4.73	0.21	4.69	34.35	9.51	0.040	252.88	12.94	51.25	99.07
	Max	37.79	1.37	15.36	68.89	43.86	0.349	501.05	83.40	94.55	337.67
	Mean	10.81	0.49	7.64	50.90	21.65	0.098	364.55	30.65	69.12	184.86
	SD	8.43	0.26	4.05	9.55	10.90	0.065	58.33	23.22	14.71	74.36
	CV/%	78.02	53.93	52.97	18.75	50.36	65.77	16.00	75.76	21.28	40.23
F2 (*n* = 18)	Min	3.31	0.16	3.77	24.25	4.77	0.018	271.60	9.28	21.50	51.57
	Max	5.95	0.80	11.77	81.39	32.29	0.098	414.15	20.32	77.55	162.77
	Mean	4.89	0.32	5.37	45.81	17.74	0.051	355.16	15.26	31.52	94.76
	SD	0.76	0.16	1.80	18.32	9.46	0.025	51.16	3.46	12.54	31.04
	CV/%	15.58	48.90	33.59	39.99	53.35	49.93	14.40	22.69	39.80	32.75
F3 (*n* = 36)	Min	0.77	0.07	1.82	9.19	7.13	0.010	150.15	6.27	10.83	17.17
	Max	9.00	2.43	7.60	54.47	34.44	0.248	470.83	29.01	66.62	193.97
	Mean	5.44	0.63	4.98	35.22	16.93	0.061	383.59	16.34	32.93	94.46
	SD	2.10	0.60	1.46	11.53	6.34	0.051	105.58	5.01	14.48	51.23
	CVs/%	38.61	94.42	29.36	32.73	37.47	83.02	19.52	30.67	43.98	49.24
Overall		7.04	0.48	5.99	43.97	18.77	0.22	367.77	20.75	44.52	124.69
Beijing [30]		4.15	0.534	-	85.7	70.3	0.204	565	42.6	67	300
Shanghai [31]		9.4	0.46	-	-	132.1	-	-	-	105.6	473.5
Guangzhou [32]		-	-	6.90	64.3	102	-	411	23.6	84.1	384
BC_soil_ [17]		4.29	0.26	8.12	30.06	12.18	0.038	311.65	12.77	16.24	34.70

## Data Availability

Some or all data and models that support the findings of this study are available from the corresponding author upon reasonable request.

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
