# Peer review of "Variation in Road Dust Heavy Metal Concentration, Pollution, and Health Risk with Distance from the Factories in a City–Industry Integration Area, China"

_ijerph, 2022, doi:10.3390/ijerph192114562_

Round 1
Reviewer 2 Report
Dear authors, in attached file you will found my comments. Kind regards, Reviewer

Round 2
Reviewer 1 Report
The manuscript can be accepted.
Author Response
Thank you for your approval, we have studied the paper carefully again and have made details correction, especially for the "Conclusions" and "Reference". Revised portion are marked in red in the paper.
Best wishes !

Reviewer 2 Report
The paper by Chen et al. represents a qualitative scientific work but have few mistakes and unanswered questions. I have few objections. Conclusion’s part needs to be more precisely written with less numbers (repetition of results) and more concluded remarks. At the end, I am wondering about references. This topic is actual in other parts of the world not only in China. I encourage authors to revise their references and put some references from other parts of the world. Therefore, I recommend acceptation of the paper with minor revision.
Author Response
Thank you very much for your comments and suggestions. We have studied the paper carefully again and have made details correction which we hope to meet with approval. Revised portion are marked in red in the paper.
Best wishes !